# Segmentation of seventy-one anatomical structures necessary for the evaluation of guideline-conform clinical target volumes in head and neck cancers

**Alexandra Walter** [1,2,3]                                  ALEXANDRA.WALTER@KIT.EDU
**Goran Stanic**[1,2,4]                                       GORAN.STANIC@DKFZ-HEIDELBERG.DE
**Philipp Hoegen**[2,5,6,7]                                   PHILIPP.HOEGEN@MED.UNI-HEIDELBERG.DE
**Sebastian Adeberg**[2,5,6,7,8,9]                            SEBASTIAN.ADEBERG@MED.UNI-HEIDELBERG.DE
**Oliver Jäkel**[1,2,9]                                       O.JAEKEL@DKFZ-HEIDELBERG.DE
**Martin Frank**[3]                                          MARTIN.FRANK@KIT.EDU
**Kristina Giske**[1,2]                                       K.GISKE@DKFZ-HEIDELBERG.DE

[1] *Department of Medical Physics in Radiation Oncology, German Cancer Research Center (DKFZ), Im Neuenheimer Feld 280, Heidelberg, Germany*

[2] *Heidelberg Institute of Radiation Oncology (HIRO) & National Center for Radiation Research in Oncology (NCRO), Heidelberg, Germany*

[3] *Karlsruhe Institute of Technology, Steinbuch Center for Computing, Hermann-von-Helmholtz-Platz 1, Eggenstein-Leopoldshafen, Germany*

[4] *Department of Physics and Astronomy, University of Heidelberg, Heidelberg, Germany*

[5] *Department of Radiation Oncology, Heidelberg University Hospital, Heidelberg, Germany*

[6] *Clinical Cooperation Unit Radiation Oncology, German Cancer Research Center (DKFZ), Heidelberg, Germany*

[7] *National Center for Tumor diseases (NCT), Heidelberg, Germany*

[8] *German Cancer Consortium (DKTK), Partner Site Heidelberg, Heidelberg, Germany*

[9] *Heidelberg Ion-Beam Therapy Center (HIT), Department of Radiation Oncology, Heidelberg University Hospital, Heidelberg, Germany*

**Editors:** Under Review for MIDL 2023

## Abstract

Expert guidelines standardize the delineation of the clinical target volume by giving neighboring anatomical structures as boundaries. In this research, we have automated the segmentation of seventy-one most mentioned anatomical structures in the expert guidelines. For most structures there are no segmentation accuracies found in the literature. For others, the DICE improves in our research. The sDICE with a 2 mm tolerance shows clinical acceptance for most structures. Overall, we set benchmarks for several structures of the head and neck anatomy. With these segmentations the guideline-conformance of clinical target volumes can be measured.

**Keywords:** Expert guidelines, automatic segmentation, anatomical structures, multi-label segmentation, target volume delineation, head and neck cancer.

## 1. Introduction

In radiation therapy the precise delineation of the target volume is crucial for tumor control. Especially in the head and neck area, in which anatomical structures are closely neighboring

each other, there is a fine line between tumor control and the sparing of healthy tissues. Consensus expert guidelines are accepted to standardize the segmentation of target volumes (Grégoire et al., 2018; Gregoire et al., 2014). The clinical target volume (CTV) comprises the tissues infiltrated by microscopic tumor cells and in the expert guidelines its outline is described by bordering anatomical structures.

Manual delineations are time-consuming and show large inter- and intra-observer variability (van der Veen et al., 2019). While deep learning methods show good results on the segmentation of organs at risk (OAR) (Nikolov et al., 2018), despite years of effort the progress of CTV segmentation is heavily affected by the inconsistent manual delineations (Cardenas et al., 2018; Strijbis et al., 2022).

Our previous work has shown that the rules and boundaries given by the expert guidelines are not learned by a neural network during supervised training. Our new approach to the automatic delineation of CTVs is the direct automated application of the rules. For that, all necessary anatomical structures need to be segmented, then the relevant boundaries can be extracted and combined based on the rules given by the expert guidelines to construct a guideline-conform CTV.

Earlier studies on the segmentation of anatomical structures have only published results on a small subset of necessary structures. We have investigated the precision to which all necessary anatomical structures can be automatically segmented.

## 2. Methods

For the study, the most important seventy-one anatomical structures mentioned in the expert guidelines were manually delineated by six trained observers with a standard operating procedure. Delineations were made for 104 calibrated planning CT scans aggregated from four different study cohorts. All patients gave informed consent. The study is approved from the ethics committee XXX. Three different nnUnet models were trained with default parameters on disjunct subsets of the structures (Isensee et al., 2021). Structures that were not present in the manual labels were neglected in the analysis.

## 3. Results

Table 1 shows the DICE and sDICE (Nikolov et al., 2018) with a 2 mm tolerance for the ten most mentioned structures in both guidelines. The DICE results are comparable or better than the ones in the literature for the sternocleidomastoid muscle ($0.73 \pm 0.02$ in Weber et al. (2021)), the pharyngeal constrictor muscles and the carotid arteries ($0.71 \pm 0.10$ and $0.68 \pm 0.11$ in Van Dijk et al. (2020)), and the pharynx ($69.3 \pm 6.3$ in Ibragimov and Xing (2017)). Only for the vertebra C1 there are better results found (approximately 0.96 in Wasserthal et al. (2022)) which are not reproducible when their method is applied to our own data ($0.83 \pm 0.03$). To our knowledge, for the DICE of all other structures as well as the sDICE for all structures, there is no segmentation accuracy published. The DICE and sDICE are comparable to the inter-observer variability calculated for a subset of representative structures delineated by two of our trained observers.

The DICE and sDICE results for different groups of anatomical structures are given in Figure 1. Groups of structures with better contrast (i.e. air, bones) show better results

Table 1: Comparison of predicted and manual labels of ten anatomical structures. Mean ± standard deviation of the DICE and sDICE with a 2 mm tolerance.

| Structure | DICE | sDICE |
|---|---|---|
| pharynx | 0.82 ± 0.08 | 0.89 ± 0.07 |
| sternal manubrium | 0.91 ± 0.06 | 0.93 ± 0.08 |
| hyoid bone | 0.82 ± 0.07 | 0.95 ± 0.06 |
| vertebra C1 | 0.86 ± 0.04 | 0.92 ± 0.04 |
| thyroid cartilage | 0.84 ± 0.08 | 0.96 ± 0.04 |
| cricoid cartilage | 0.67 ± 0.16 | 0.80 ± 0.14 |
| sternocleidomastoid muscle | 0.81 ± 0.12 | 0.88 ± 0.12 |
| scalenus muscle | 0.72 ± 0.10 | 0.84 ± 0.10 |
| thyro-hyoid muscle | 0.51 ± 0.19 | 0.84 ± 0.17 |
| pharyngeal constrictor muscles | 0.60 ± 0.14 | 0.81 ± 0.10 |
| common carotid artery | 0.79 ± 0.11 | 0.92 ± 0.08 |
| internal carotid artery | 0.57 ± 0.19 | 0.77 ± 0.19 |

than the other groups. The largest variation in both coefficients is shown in the largest and most diverse group, the muscles, as well as in the smallest group summarizing the results of only two cartilages.

Since the DICE is by design more sensitive to deviations in small structures, the lowest DICE scores in each group are scored by narrow structures. Except for the right internal carotid artery, the sDICE of those structures does not show any saliencies. Because of inconsistent labeling, the tonsils were excluded from the analysis.

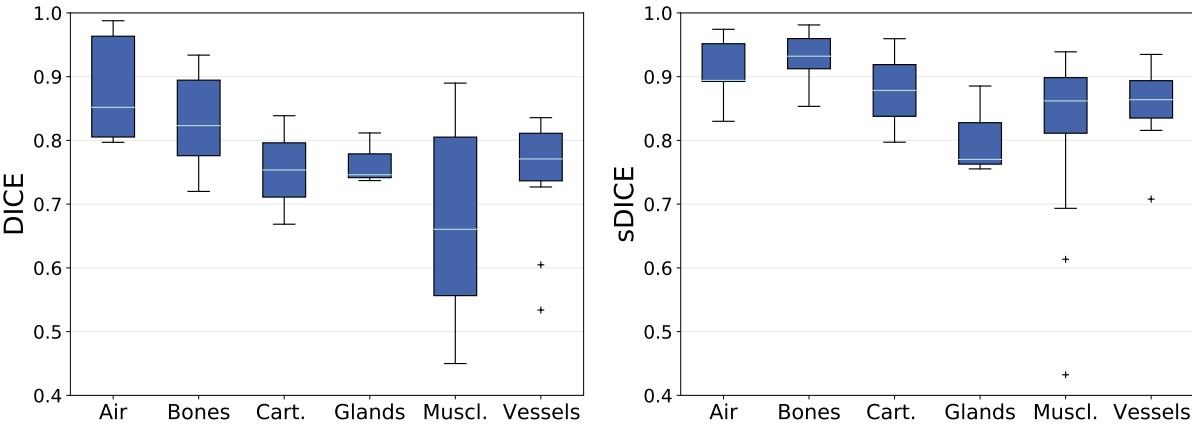

Figure 1: Comparison of predicted and manual labels between groups of anatomical structures with DICE and sDICE with 2 mm tolerance. Quantities per group were Air (6), Bones (11), Cartilages (2), Glands (3), Muscles (26), Vessels (11).

## 4. Discussion

The question whether the segmentations are accurate enough to evaluate guideline-conformance of a CTV is answered by the predominantly large values for the sDICE with 2 mm tolerance. In radiation therapy a deviation of 2 mm or less is considered acceptable for clinical use. One limitation is the inconsistent delineation of the tonsils and thus, their exclusion from further use.

Overall, this research establishes the first ever segmentation results for several structures of the head and neck anatomy, improves the segmentation results of other structures and collects all the results in one paper. These results will be a benchmark for following research on automatic segmentation in the head and neck area.

We can now automatically measure the degree of guideline-conformance of each single CTV delineation by analyzing the overlap between a CTV and the anatomical structures that should be excluded from the CTV or included in the CTV with respect to the expert guidelines. Identifying critical cases will support clinicians in delivering more consistent CTV delineations for radiotherapy planning.

## Acknowledgments

We thank Susanne Labudek, Ishan Echampati, Dorothee Kahn, Ilsa Beig, Woojin Choi and Samira Hiller for manually segmenting the anatomical structures on the planning CT scans and discussing the neck node levels. Further, we thank the Helmholtz Information & Data Science School for Health for funding AW.

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
