# OpenReview forum: "Segmentation of seventy-one anatomical structures necessary for the evaluation of guideline-conform clinical target volumes in head and neck cancers"
_MIDL.io/2023/Short_Paper_Track — MIDL 2023 Short paper track Poster_

### Official Review · Reviewer_7sdN · 2023-04-15
**strong results, clinically useful problem**

**Rating:** 8
**Confidence:** 4

**Review:**

nice paper, clear exposition

strong results

probably has good clinically useful applications

---

### Official Review · Reviewer_g98P · 2023-04-24

**Rating:** 6
**Confidence:** 4

**Review:**

This paper first delineated the most important 71 anatomical structures by 6 trained annotators, and then trained 3 different nnUnet models to segment these 71 target structures. The experimental evaluations show dice improvement by using these fine-grained annotated data. However, there are several questions regarding this paper: (1) can you provide more information regarding the 104 CT scans? (2) why not report the mean DSC for all structures? (3) the author mentioned that three nnUnet were used, but it is unclear what was the difference between them and which nnUnet that generated the results in Table 1?  (4) Will the author open-source the dataset?